# Dependency Structures in Cryptocurrency Market from High to Low Frequency

**DOI:** 10.3390/e24111548

**Published:** 2022-10-28

**Authors:** Antonio Briola, Tomaso Aste

**Affiliations:** 1Department of Computer Science, University College London, London WC1E 6BT, UK; 2Center for Blockchain Technologies, University College London, London WC1E 6BT, UK; 3Systemic Risk Center, London School of Economics, London WC2A 2AE, UK

**Keywords:** complex systems, network science, econophysics, economics, financial markets, cryptocurrencies

## Abstract

We investigate logarithmic price returns cross-correlations at different time horizons for a set of 25 liquid cryptocurrencies traded on the FTX digital currency exchange. We study how the structure of the Minimum Spanning Tree (MST) and the Triangulated Maximally Filtered Graph (TMFG) evolve from high (15 s) to low (1 day) frequency time resolutions. For each horizon, we test the stability, statistical significance and economic meaningfulness of the networks. Results give a deep insight into the evolutionary process of the time dependent hierarchical organization of the system under analysis. A decrease in correlation between pairs of cryptocurrencies is observed for finer time sampling resolutions. A growing structure emerges for coarser ones, highlighting multiple changes in the hierarchical reference role played by mainstream cryptocurrencies. This effect is studied both in its pairwise realizations and intra-sector ones.

## 1. Introduction

Financial markets are complex systems [1]. The main source of complexity comes from the intricate interaction of heterogeneous actors following various strategies designed to impact at different time scales. They are highly stochastic environments with a low signal to noise ratio, dominated by strong non-stationary dynamics and characterized by feedback loops and non-linear effects [2,3,4]. Despite their complexity, financial systems are governed by a rather stable and partially identified framework of rules [5]. This last characteristic, jointly with the possibility to continuously monitor them across time, makes financial systems well suited for statistical characterization [6] and a good playground for the study of complex systems in general. In this paper, we analyse the behaviour of cryptocurrency market. A cryptocurrency is defined as a digital instrument for value transfer that exploits cryptography and distributed ledgers for security and decentralization [7]. As currencies, they have properties similar to fiat currencies [8]. The main differences being the exclusion of financial institutions as intermediaries [9] and not being controlled and regulated by any central authority [10]. Thanks to the above mentioned characteristics, cryptocurrency market is available 24 h a day, 7 days a week, and transactions take place between individuals with different physical locations across the globe [8]. Standard features of financial systems joined with peculiarities listed above, make cryptocurrencies highly volatile instruments. Finding assets with similar behaviours responding to endogenous or exogenous events is, hence, a challenging, but extremely valuable, exercise both from theoretical and applicative perspectives (e.g., risk management and investment). The ready access availability of large volumes of market data ease research on these instruments with respect to classical financial ones. Indeed, one of the main limits faced by research in the field of financial applications is the lack of easy access and share of high-quality data. In most cases they are sensible data, owned and managed by private financial institutions. Cryptocurrencies are traded on digital currency exchanges (DCEs) which, differently from traditional exchanges, allow to easily access both online and historical data. Exploiting this and using instruments provided by network science, one can successfully build models able to capture and describe individual and collective behaviours in cryptocurrency market.

A network (or graph) represents components of a system as nodes (or vertices) and interactions among them as links (or edges). The number of nodes defines the size of the network. The number of links defines the sparsity (or, conversely, density) of the network. Reversible interactions between components are represented through undirected links, while non-reversible interactions are represented as directed links. Networks have been successfully used in many application domains. Some examples are social networks [11,12], security [13], epidemiology [14,15], neuroscience [16], drug design [17], management [18], and economic forecasting and modelling [6,19,20,21,22,23,24,25]. Many of the above cited works share the peculiarity to study networks with a size varying as a function of time. In the current research work, on the contrary, we focus on networks with a fixed size. Starting from a set of 25 liquid cryptocurrencies, we exploit the power of state-of-the-art network-based information filtering approaches (i.e., MST [26] and TMFG [27]) to build robust models capturing strong interactions among assets and pruning, at the same time, weakest ones. We hence investigate dependency structures of the networks at 6 different time horizons spanning from 15 s to 1 day. For each time horizon, we test the stability, statistical significance, and economic meaningfulness of the graphs. Such a research effort is led by two main motivations. The first one is the will to describe core dependency structures of the cryptocurrency market in a systematic way, providing a detailed characterization of the reference role played both by mainstream cryptocurrencies and by peripheral ones. The second is related to the possibility to do this at a wide range of time scales including intra-minute resolutions. Such a characterization is relevant for many reasons. Cryptocurrency market is affected by daily changes related to the introduction of new coins, collapse of existing ones, updates on existing protocols, etc. Having a stable framework able to robustly handle this intrinsic mutability, highly eases investment, and risk management decisions and provide a ductile instrument for research purposes. Such a framework should be also able to handle dynamics of cryptocurrencies showing similar characteristics and behaviours (i.e., belonging to the same sector). Dependency structures are, hence, investigated, both at an intra-sector and pairwise level. Unfortunately, there is no consensus on a unique mapping between cryptocurrencies and sectors. We adopt the taxonomy proposed by Kraken [28] digital currency exchange. Results give a deep insight into the evolutionary process of the time dependent hierarchical organization of the chosen system of cryptocurrencies. As a further step toward robustness, we compare our results with the ones achieved in the past 20 years of similar research in the field of stock market, uncovering comparable behaviours between the two systems. From an economic and financial perspective, the study of dependency structures among cryptocurrencies at different time-scales is relevant both from a theoretical and an applicative point of view. In the first case, comparing properties of time dependent hierarchical organization of the cryptocurrency market (a relatively young market) with the ones of the equity market (a consolidated market), (i) allows to measure its degree of maturity (ii) keeping track, at the same time, of the main evolutionary phases. In the second case, such an analysis is useful as a support instrument toward the achievement of different goals spanning from portfolio construction tasks (e.g., diversification purposes) to development of multi-assets trading strategies acting at different time-scales. Our contribution to the existing literature is threefold: (i) we are the first to use TMFG as an information filtering approach to model dependency structures among crypto-assets, (ii) we propose a rigorous network-based study of cryptocurrency market allowing to compare emerging dynamics to the ones observed on traditional financial markets (e.g., the “Epps effect”), and (iii) we are the first to describe the evolution of dependency structures among cryptocurrencies at time scales spanning from intra-minute to daily resolution.

The rest of the paper is organised as follows. In Section 2, we review the previous research on applications of network science to financial systems modelling. In Section 3.1, we discuss the data acquisition and transformation pipeline. In Section 3.3, Section 3.4, and in Section 3.5, we characterise cross-correlation between cryptocurrencies as a measure of similarity and dependency. We show how to obtain a dissimilarity measure based on cross-correlation and we review the building process and properties of MSTs, PMFGs and TMFGs. In Section 4, we present results obtained applying methods reported in Section 3. In Section 5, we conclude by discussing the economic and financial interpretation of our findings.

## 2. Related Work

Networks have been extensively used in order to model economic and financial systems. The work by [19] can be identified as a foundational one. It demonstrates the possibility to find a hierarchical arrangement of stocks traded in a financial market by investigating the daily time series of logarithmic price returns. A graph is obtained, exploiting information contained in the correlation matrix computed between all pairs of stocks of the portfolio by considering the synchronous time evolution of the logarithmic returns. Building on the work by [19], the paper by [29] shows that sets of stock index time series can be used to extract meaningful information about the links between different economies across the world. This goal is successfully achieved provided that the effects of the non-synchronous nature of the time series and of the different currencies used to compute the indices are properly taken into account. The work by [22] further extends the research by [19], studying modifications of the hierarchical organization of a set of stocks switching from high- to low-frequency time scales. As a first step, authors report a decrease in correlation between pairs of assets switching from coarser to finer time sampling resolutions. Such a phenomenon is known as “Epps effect” [30]. This analysis is extended, investigating both pairwise and intra-sector dynamics. They show the emergence of a more complex network structure at coarser time sampling resolutions, highlighting multiple changes in the hierarchical reference role played by sectors’ representative assets. The work by [20] tests the robustness of the findings of the previously cited research works for longer periods of investigation and demonstrates that networks describing the financial domain cannot be reproduced by a random market model [31,32] and by the one-factor model [33]. Such results are also investigated in [21] which specifically shows how the topology of the networks in financial systems can be used to validate or falsify simple, although widespread, market models. This work also extends the previously cited ones introducing an analysis of the networks built on the volatility of financial time series. More recently, the work by [6] shows vulnerabilities of MST [26] in representing complex systems and proposes the usage of a planar graph, the PMFG [34], as an alternative. This research work also presents a set of methods to validate the statistical significance and robustness of achieved empirical results. The centrality role of specific financial sectors is finally investigated and the evolution of the Financial sector as a reference one is analysed over a period of 10 years. Recently, some of the network-based information filtering approaches have been sparsely applied to the cryptocurrency market. Results consistent with the ones described in our paper have been recently described by [35], adopting a different methodology. In this research, exploiting the index cohesive force [36], the author describes the changes in the hierarchical order of the most influential cryptocurrencies over a period of five years. He shows how Ethereum gradually becomes the most influential cryptocurrency at the detriment of Bitcoin. It is also useful to mention the work by [37], where, for the first time, the authors suggest a network-based approach to study the interdependencies between log-returns of cryptocurrencies, with a special focus on Bitcoin. They use the MST method in order to group assets into hierarchical clusters and they highlight the potential existence of topological properties of the cryptocurrency market. This work is extended by [38], where, the authors adopt the MST and the PMFG to study the change in cryptocurrency market’s network structure before and after the COVID-19 outbreak. The last work to be mentioned is the one by [39], where the author points out how most of the studies on cryptocurrency market are focused only on daily data without considering other options. Using a range of frequencies spanning from one minute to weekly data, he shows how it is possible to detect different profitable frequencies and underlines the relevance of analysing frequencies different from daily ones.

## 3. Methods

### 3.1. Data

The vast majority of digital currency exchanges provide a free Rest API (or a Web-socket) allowing users to access both historical OHLCV (open, high, low, close, volume) data and online Limit Order Book- and trades-related data. In addition to this, there is a growing number of services providing out of the box, unified APIs which support many exchanges and merchant APIs. The work by [8] reports a comprehensive and detailed overview of the services currently available for data retrieving. In the current work, we use data from the FTX [40] digital currency exchange. They are entirely accessed through the CCXT [41] Python package. We use OHLCV data for 25 cryptocurrencies (see Table 1) sampled at time horizons Δt∈[15 s, 1 min, 15 min, 1 h, 4 h, 1 day]. For each time horizon, a sample can be defined as a “time bin”. Opening and closing prices are, respectively, the first and the last price of the time bin, high and low price are, respectively, the highest and the lowest price of the time bin and can technically happen in any order, and the volume is is defined as the sum of the volumes traded in the time bin. In the rest of the paper, we will use a second-based definition of time horizons. This means that we will refer them as Δt∈[15, 60, 900, 3600, 14,400, 86,400]. Qualitatively, we will often speak about finer and coarser time horizons. In the first case, we want to indicate elements nearer to the lower bound of the set of time sampling resolutions, while, in second case, we want to indicate elements nearer to the upper bound of the set of time sampling resolutions.

All the considered cryptocurrencies are liquid with a medium-to-high market capitalization. An exception is Cream, which has a low capitalisation. The only constraint in the selection process of cryptocurrencies is their historical availability on the FTX digital currency exchange. Indeed, it is worth noting that each digital currency exchange allows to access historical data only starting from the date a specific asset has been quoted on the exchange itself. The period under analysis spans between 1 January 2021 to 28 February 2022. Despite the high-quality of data, rare missing values are detected at the finest time sampling resolution (i.e., Δt=15). In this case, they are filled using the nearest valid observation. Logarithmic returns (named in the rest of the paper as log-returns) *x* of closing prices *p* at time *t* for a given cryptocurrency *c*, are computed as follows:(1)$$x_{c}\left(t\right)=\log\left(p_{c}\left(t\right)\right)-\log\left(p_{c}\left(t-\Delta t\right)\right).$$

The assumption of returns’ stationarity is validated for each $x_{c}\left(t\right)$ through the Augmented Dickey Fuller (ADF) [43] test.

### 3.2. Correlation-Based Filtering

Understanding how variables evolve, influencing the collective behaviour, and how the resulting system influences single variables is one of the most challenging problems in complex systems. In order to extract such an information from the set of synchronous time series discussed in Section 3.1, we proceed by determining their Pearson’s correlation coefficient at each time horizon Δt. The Pearson’s estimator of the correlation coefficient, for non-overlapping increments, between two synchronous data series with length TΔt is:(2)$$\rho_{i,j}\left(\Delta t\right)=\frac{\frac{1}{T}\sum_{u=1}^{T}\left(x_{i}\left(u\Delta t\right)-\mu_{i}\right)\left(x_{j}\left(u\Delta t\right)-\mu_{j}\right)}{\sigma_{i}\sigma_{j}}$$
where $\mu_{i(j)}$ and $\sigma_{i(j)}$ are, respectively, the sample mean and the sample standard deviation of the data series $x_{i(j)}\left(t\right)$. The Pearson’s correlation coefficient is a widespread measure efficient at catching similarities between the evolution process of financial assets’ prices [6]. By definition, $\rho_{i,j}\left(\Delta t\right)$ has values between −1 (meaning that the two synchronous time series are completely, linearly anti-correlated) and +1 (meaning that the two synchronous time series are completely, linearly correlated). When $\rho_{i,j}\left(\Delta t\right)=0$, the two synchronous time series are linearly uncorrelated. The correlation matrix C is n×n (where *n* is the number of variables) symmetric, with elements on the diagonal equal to one (i.e., $\rho_{i,i}\left(\Delta t\right)=1$). For each time horizon Δt, n(n−1)/2 correlation coefficients completely characterize the correlation matrix. From a network science perspective, the correlation matrix can be considered as a fully connected graph where each asset is represented by a node and each pair of assets is joined by an undirected edge representing their correlation.

### 3.3. Minimum Spanning Tree (MST)

Based on the correlation matrix, we want to build an undirected graph whose topology captures dependency structures among cryptocurrencies’ log-returns time series and that is greatly reduced in the number of edges with respect to a complete graph. In such a network, all the relevant relations must be represented. At the same time, the network should be kept as simple as possible. The simplest connected graph is a spanning tree. Minimum spanning trees (MSTs) [26] are largely used in multivariate analysis; they represent a class of networks that connect all the vertices without forming cycles (i.e., closed paths of at least three nodes). MSTs are often computed with respect to a distance metric, so that minimizing the metric corresponds to linking assets that are close to each other. As a product of their building process, MSTs retain the maximum possible number of distances [19] minimizing, at the same time, the total edge distance. In [19], MSTs are computed using the Euclidean distance [44]:(3)$$d_{i,j}=\sqrt{2(1-\rho_{i,j})}.$$

This definition is however too restrictive disfavouring negatively correlated variables that are equally important as the positive ones for the representation of the dependency structure [45]. In order to mitigate this limitation, we use the power dissimilarity measure:(4)$$d_{i,j}=1-\rho_{i,j}^{2}$$

The work [46] provides a complete pedagogical exposition of the determination of the MST in the context of synchronous financial time series. A general approach to the construction of the MST is to connect the less dissimilar vertices while constraining the graph to be a tree as follows:Make an ordered list of edges i,j, ranking them by increasing dissimilarity (first the edge expressing the highest similarity and last the edge expressing the highest dissimilarity).Pop the first element of the ordered list and add it to the spanning tree.If the added edge creates a cycle then remove the edge, otherwise skip to step 4.Iterate the process from step 2 until all pairs have been exhausted.

Such an algorithm for the construction of the MST is known as the Prim’s algorithm [47]. The resulting network has n−1 edges. Considering that the system of cryptocurrencies analysed in the current paper is made of n=25 assets (i.e., nodes), the resulting MST contains 24 edges (the code used to compute MSTs can be retrieved at https://github.com/shazzzm/topcorr; last access on 27 October 2022).

### 3.4. Planar Maximally Filtered Graph (PMFG)

The MST is a powerful method to capture meaningful relationships in a network structure describing a complex system. However, this method presents some aspects that can be unsatisfactory. The main constraint is that it has to be a tree (i.e., it cannot contain cycles). This characteristic makes impossible to represent relationships among more than two variables showing strongly correlated behaviours in their dynamics. In order to maintain the same powerful filtering properties of the MST and adding, at the same time, extra links, cycles, and cliques (i.e., complete subgraphs) in a controlled manner, it has been proposed to use the Planar Maximally Filtered Graph (PMFG) [48,49,50,51]. PMFG can be viewed as the first incremental step towards complexity after the MST. Indeed, instead of being a tree, the algorithm impose planarity. A graph is said to be planar if it can be embedded in a sphere without edges crossing. The foundational work by [6] provides a comprehensive pedagogical exposition of the determination of the PMFG. A general approach to the construction of the PMFG can be resumed as follows:Make an ordered list of edges i,j, ranking them by increasing dissimilarity (first the edge expressing the highest similarity and last the edge expressing the highest dissimilarity).Pop the first element of the ordered list and add it to the graph.If the resulting graph is not planar, then remove the edge, otherwise skip to step 4.Iterate the process from step 2 until all pairs have been exhausted.

It has been proved that the MST is always a sub-graph of the PMFG [48]. PMFG has 3×(n−2) edges and a number of 3-cliques larger or equal to 2n−4. We remark that also 4-cliques can be present in this kind of graph.

### 3.5. Triangulated Maximally Filtered Graph (TMFG)

The PMFG presents two main limits: it is computational costly and it is a non-chordal graph. A graph is said to be chordal if all cycles made of four or more vertices have a chord which reduces the cycle to a set of triangles. A chord is defined as an edge that is not part of the cycle but connects two vertices of the cycle itself. In order to bypass these two constraints, the Triangulated Maximally Filtered Graph (TMFG) [27] has been proposed. A general approach to the construction of the TMFG can be resumed as follows:Make an ordered list of edges i,j, ranking them by increasing dissimilarity (first the edge expressing the highest similarity and last the edge expressing the highest dissimilarity).Find the 4 nodes with the lowest sum of edge weights with all other nodes in the graph and connect them forming a tetrahedron with 4 triangular faces.Identify and add the node that minimize the sum of its connections to a triangle face already included in the graph, forming three new triangular faces.If the graph reaches a number of edges equal to 3n−6, then stop, otherwise go to step 3.

Such an algorithm extracts a planar subgraph which optimises an objective function quantifying the gain of adding a new vertex to the existing tetrahedron. Compared to the PMFG, the TMFG is more efficient to be computed and is a chordal graph. The chordal structural form allows to use the filtered graph for probabilistic modeling [52,53]. A TMFG has 3×(n−2) edges (with *n* representing the number of nodes) and contains both 3-cliques and 4-cliques. Considering that the system of cryptocurrencies analysed in the current paper is made of n=25 assets (i.e., nodes), the resulting TMFG contains 69 edges, 88 3-cliques, and 22 4-cliques (The code used to compute TMFGs can be retrieved at https://github.com/shazzzm/topcorr; last access on 27 October 2022.).

## 4. Results

Figure 1a and Figure 2a report the MST and the TMFG computed at horizon Δt=15. Figure 1b and Figure 2b report the MST and the TMFG computed at horizon Δt=86,400. Full set of MSTs computed following the procedure described in Section 3.3 is reported in Appendix A. Full set of TMFGs computed following the procedure described in Section 3.5 is reported in Appendix B.

As a preliminary step into the study of the information level carried by the two network-based information filtering approaches, Figure 3 shows how pairwise (see Figure 3a) and the intra-sector (see Figure 3b) average Pearson’s correlation coefficient 〈ρ〉 evolves as a function of time horizon Δt. Figure 3a reports the mean Pearson’s correlation coefficient computed averaging over the n(n−1)/2=300 off-diagonal elements of the whole correlation matrix **C** at different time horizons. In order to give a more comprehensive view of the evolutionary dynamics of the mean pairwise correlation coefficient, we also report three meaningful percentile intervals. We observe that the average correlation coefficient 〈ρ〉 increases with time horizon Δt from a value equals to 0.19 at Δt=15 to a value equals to 0.47 at Δt=86,400. The value at Δt=15 corresponds to the minimum average correlation coefficient across time horizons. On the other hand, the maximum average correlation coefficient does not coincide with the one computed at the maximum time horizon. It is instead detected at horizon Δt=14,400, which corresponds to an intra-day resolution (i.e., 4 h). On average, the most prominent pairwise correlation weakenings are observed for most correlated pair of assets (i.e., those pairs of cryptocurrencies having a correlation coefficient included into highest percentiles).

Figure 3b reports mean Pearson’s correlation coefficient computed averaging over the $n_{s}\left(n_{s}-1\right)/2$ correlation coefficients of the $n_{s}$ assets belonging to one specific sector [42] at different time horizons. Specifically, we report dynamics for Currencies, Smart Contracts, and Centralized Exchanges sectors. This choice is completed considering the relevance of the three sectors. The relevance of sectors is defined in relation to results discussed later in this section. An intra-sector scenario shows trends comparable to the ones observed in pairwise context. All the previously discussed dynamics are here more pronounced. In both cases, we observe the “Epps effect”, i.e., a decrease in pair correlations at finer time sampling resolutions. This effect has been extensively studied in equity markets by [22,30]. Results reported in Figure 3 show how, also in the cryptocurrency market, the intra-sector correlation increases faster than pairwise one. The “Epps effect” is, hence, more pronounced within each sector than outside it. Going deeper, in Appendix C, we compare the probability distribution of correlation coefficients in the empirical correlation matrix **C** with the probability distribution of correlation coefficients filtered, respectively, by the MST and by the TMFG at different time horizons. We also report the probability distribution of correlation coefficients for surrogate multivariate time series obtained by randomly shuffling log-returns time series of the 25 cryptocurrencies listed in Table 1. This step is performed in order to evaluate the null hypothesis of uncorrelated returns for the considered portfolio of cryptocurrencies. Results give us the possibility to asses the statistical significance of average correlation coefficients chosen both by MST and by TMFG networks. These findings are reported in a synthetic way in Table 2. The extended count and the corresponding statistical meaning of links having a value higher than the minimum and lower than the maximum correlation coefficient detected by shuffling log-returns time series at different time horizons for the three scenarios are reported in Appendix D.

Average correlation coefficients for MSTs and TMFGs are always greater than the ones computed on the empirical correlation matrix **C**. The difference between cross-horizons mean of average correlation coefficients filtered by MSTs and cross-horizons mean of average correlation coefficients in **C**, is equal to 0.16. The difference between cross-horizons mean of average correlation coefficients filtered by TMFGs and cross-horizons mean of average correlation coefficients in **C**, is equal to 0.12. Correlation coefficients filtered by TMFGs are always lower than the ones filtered by MSTs. This depends on the fact that, as reported in Section 3.5, the TMFG contains, by construction, more information than the MST. The mean difference between average correlation coefficients filtered by MSTs and the ones filtered by TMFGs, is equal to 0.03. Results reported in Table 2 confirm that the two filtering approaches prune weakest correlations among considered cryptocurrencies keeping only the strongest ones. Differently from what happens for the empirical correlation matrix **C**, results for both MST and TMFG are always statistical significant across time horizons. These results enforce the evidence that both MST and TMFG carry information about strongest interactions observed in the system, disregarding most of the links consistent with the null hypothesis of uncorrelated data. It is worth noting that such an analysis does not tell much about the statistical robustness of links selected by the two network-based information filtering approaches. In order to perform such an investigation, we adopt the technique proposed by [54]. For each time horizon Δt, we sample 1000 bootstrap replicas r=1,…,1000 of the empirical log-returns time series data. The length of empirical data and the one of each replica is kept equal. We compute the MST*(*r*) and the TMFG*(*r*) for each replica *r*. For each time sampling resolution, we map each link of the original MST and TMFG to an integer number and we count the number of links present both in the MST and TMFG and in each of the MST*(*r*) and TMFG*(*r*). Table 3 reports, for each time sampling interval Δt, the number of links of the empirical MST and TMFG with a bootstrap value larger than 95%.

Results in Table 3 show how, in the case of the MST, the robustness of the underlying network structure decreases for coarser time sampling resolutions. A consistent result has been observed by [50] in equity markets. This finding can be explained in two different ways. The first and most straightforward explanation is the statistical one and can be resumed as follows: the higher number of samples at finer time sampling resolutions implies higher statistical significance, while the lower number of samples at coarser time sampling resolutions imply lower statistical significance. A second explanation can be given looking at the structure of the networks reported in Appendix A. At finer time sampling resolutions, we observe less structured networks where numerous small-degree nodes (spokes) coexist with few anchor ones (hubs) characterised by an exceptionally high number of links. At coarser time sampling resolutions we observe more structured networks with a less imbalanced degree distribution. Such a topological change directly implies a loss in the links’ statistical robustness. The case of TMFG is different. Statistical robustness of the network is maintained across horizons without significant draw-downs. Indeed, during the optimization phase of the objective function, TMFG tends to be marginally exposed to local minima, being robust to dramatic topological changes.

These last findings can be formally characterised studying the evolution of the average shortest path in MST and in TMFG as a function of time sampling resolution. Figure 4 reports the significant different behaviour in compactness’ evolutionary dynamics of the two network-based information filtering approaches. In the case of MST, the minimum length of the average shortest path is equal to 2.46 and is detected at Δt=15, while the maximum length is equal to 3.05 and is detected at Δt=86,400. In the case of TMFG, we observe a strong compactness across time horizons. The minimum length of the average shortest path is equal to 1.83 and is detected at Δt=3600, while the maximum length is equal to 1.9 at Δt=60. In the case of MST, at the finest time sampling resolution (i.e., Δt=15), we observe a structurally simple network with two cryptocurrencies (i.e., Ethereum and Bitcoin) acting as a hierarchical reference for the majority of other assets. This topological structure persists switching to time horizon Δt=60. Several changes in nodes’ reference roles can be observed for networks sampled at time horizons Δt=900 and Δt=3600.

In both cases Ethereum maintains its reference role even reducing its centrality. Bitcoin, on the contrary, is gradually replaced in its role by Litecoin and FTX token (both part of the Bitcoin’s cluster at time horizon Δt=60). This structural transition is evident at Δt=14,400 and fully realised at Δt=86,400.

In the case of TMFG representations, it is harder to graphically detect similar dynamics. Figure 5 offers a comparative perspective between behaviours of the two network-based information filtering approaches. It shows horizon dependent evolutionary dynamics of degree centrality (i.e., measurement of the number of connections owned by a node) [55,56,57] for Ethereum, Bitcoin, Litecoin, and FTX Token both for MST and for TMFG. Cross-assets similarities can be detected between the two types of graphs. In the case of MSTs, degree centrality is less sensitive to minor changes in reference roles played by mainstream cryptocurrencies across time horizons, amplifying only ‘extreme’ ones. In the case of TMFGs, on the contrary, the same centrality measure is able to capture even small variations in network structure. This observation can be easily explained considering the amount of information the two representations are able to express. This study can be further extended looking at sectors of cryptocurrencies instead of at singular assets. Figure 6 reports the evolution of degree centrality for the three sectors Ethereum, Bitcoin, Litecoin, and FTX Token belong to: the Currencies sector, the Smart Contracts sector, and the Centralized Exchanges sector. We remark that there is no consensus on a unique mapping between cryptocurrencies and sectors. The taxonomy adopted in the current paper is described in [42] and corresponds to the one used by Kraken [28].

Figure 6a shows how, in the case of MST, the average degree centrality for the Smart Contracts sector strongly decreases starting from time horizon Δt=3600, following the trend of its leading representative: Ethereum cryptocurrency. The Currencies sector, on the other hand, does not experience a decreasing trend and tends to remain stable across time horizons with low level of oscillations. In this case the loss of centrality of Bitcoin after time horizon Δt=900, is immediately compensated by Litecoin, which reaches a hierarchical reference role at coarser time sampling resolutions. The case of Centralized Exchanges sector is different. It is stable across time horizons, without experiencing any change in intra-sector reference role dynamics and always following the behaviour of its main representative, FTX token (see Figure 5a). This last finding can be explained considering the source of the data used in the current research work. As explained in Section 3.1, we fetch data from the FTX digital currency exchange. This can cause, on the one hand an over-estimation of the role played by the exchange specific token, FTX Token, in the whole ecology of the system under investigation, and, on the other hand, can give a potentially biased stability to the sector the asset belongs to.

## 5. Conclusions

We investigate how cryptocurrency market’s dependency structures evolve passing from high to low frequency time sampling resolutions. Starting from the log-returns of 25 liquid cryptocurrencies traded on the FTX digital currency exchange at 6 different time horizons spanning from 15 s to 1 day, we investigate pairwise correlations demonstrating that cryptocurrency market has an “Epps effect” which is comparable to the one widely studied in the equity market. Indeed, we show that the average correlation among assets increases moving from high to low frequency time horizons and we demonstrate how this dynamic is even more evident grouping cryptocurrencies into sectors. Using the concept of power dissimilarity measure, we review the building process of two network-based information filtering approaches: MST and TMFG. If, on the one hand, MST has been historically used in the description of dependency structures of different financial markets, on the other hand, this is the very first time TMFG is used to study interactions between digital assets at different time scales. Studying topologies of MSTs at finer time sampling resolutions, we observe structurally simpler networks characterised by an hub-and-spoke configuration with statistically robust links. We observe an increase in the complexity of the networks’ shape for coarser time sampling resolutions with a decrease in links’ statistical robustness. Such an horizon-dependent structural change is reflected by the average path length of the networks, characterised by an increasing trend moving from high to low frequencies. TMFG offers a different perspective for the same problem. In this case, we do not observe dramatic changes in networks’ topologies across time horizons. Graphs are more compact and statistical robustness of links is maintained across time with negligible oscillations. As a consequence of this, the average path length is lower and almost constant across time horizons. Studying the relative position of assets in both MSTs and TMFGs through the usage of degree centrality measure, we outline the presence of multiple changes in the hierarchical reference role among the considered set of cryptocurrencies. These changes strongly characterise singular cryptocurrencies. We find that Ethereum acts as a hierarchical reference node for the majority of other assets and maintains this role across time, gradually losing its centrality at coarser time horizons. There is not a clear economic explanation for this result. We know that lots of other cryptocurrencies are based on the Ethereum’s blockchain technology but we do not think this represents a sufficient explanation to our finding. Other cryptocurrencies play a similar role with respect to smaller clusters of assets at specific time horizons. We refer specifically to Bitcoin, Litecoin, and FTX Token. Differently from Ethereum, their role does not emerge at finer time sampling resolutions and should be considered as the result of a structured evolutionary process across time horizons. We conclude stating that sectors’ dynamics captured by the chosen network-based information filtering approaches are poorly affected by the ones of their main representatives, efficiently absorbing horizon-dependent changes in cryptocurrencies dynamics. This is true especially for TMFG. Indeed, looking at the evolution of the degree centrality of the Smart Contracts and Currencies sectors, one can observe that dynamics captured by MST are strongly influenced by the ones of Ethereum and Bitcoin. This does not happen in the case of TMFG where sectors’ dynamics are typically detached from the ones of specific cryptocurrencies.

## Figures and Tables

**Figure 1 entropy-24-01548-f001:**
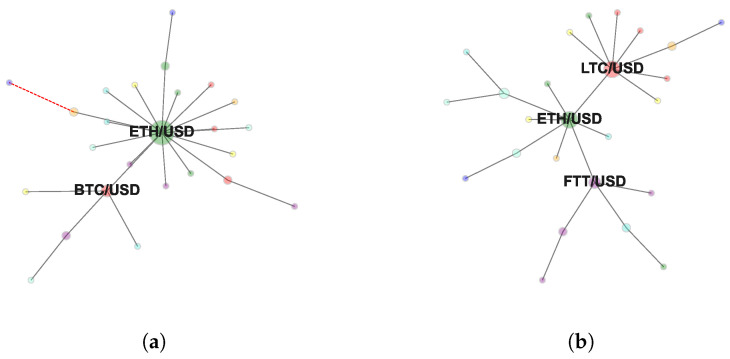
Minimum Spanning Tree representing log-returns time series’ dependency structure computed at (**a**) 15 s and (**b**) 1 day. Only hub nodes are labelled. The adopted colour mapping scheme follows the sectors’ taxonomy by [42] (see Appendix A).

**Figure 2 entropy-24-01548-f002:**
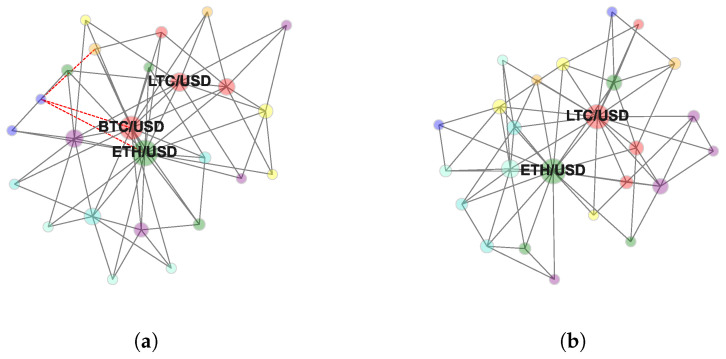
Triangulated Maximally Filtered graphs representing log-returns time series’ dependency structure computed at (**a**) 15 s and (**b**) 1 day. Only hub nodes are labelled. The adopted colour mapping scheme follows the sectors’ taxonomy by [42] (see Appendix B).

**Figure 3 entropy-24-01548-f003:**
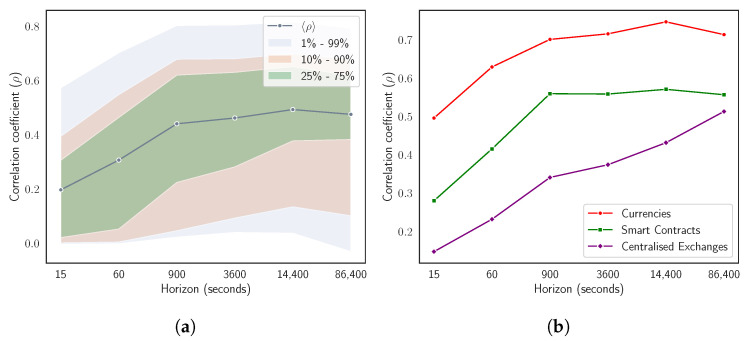
Evolutionary dynamics of the average correlation coefficient as a function of the time horizon Δt. (**a**) reports the horizon-related mean Pearson’s correlation coefficient and three meaningful percentiles computed averaging over the n(n−1)/2=300 off-diagonal elements of the whole correlation matrix **C**. (**b**) reports the horizon related mean Pearson’s correlation coefficients computed averaging over the $n_{s}\left(n_{s}-1\right)/2$ correlation coefficients of the $n_{s}$ assets belonging to one of three of the most relevant sectors defined by [42]: Currencies, Smart Contracts, Centralised Exchange sectors.

**Figure 4 entropy-24-01548-f004:**
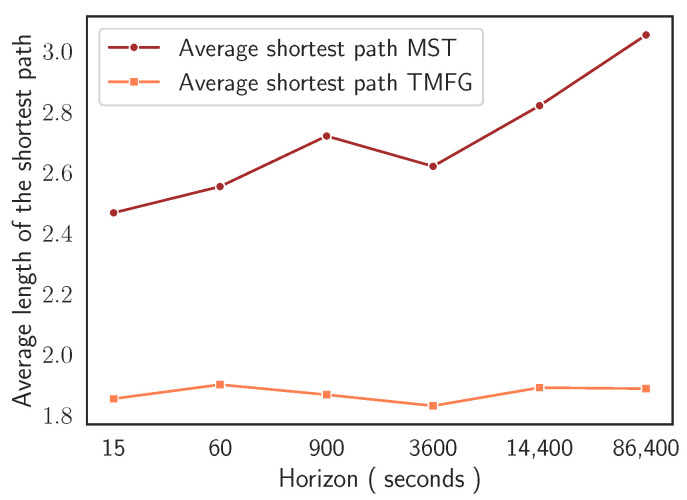
Average length of the shortest path in MST and TMFG as function of the time horizon at which log-returns are computed. We observe a decreasing compactness of MST networks at coarser time sampling resolutions. Instead, the compactness of the TMFG turns out to be stable across time horizons with low oscillations.

**Figure 5 entropy-24-01548-f005:**
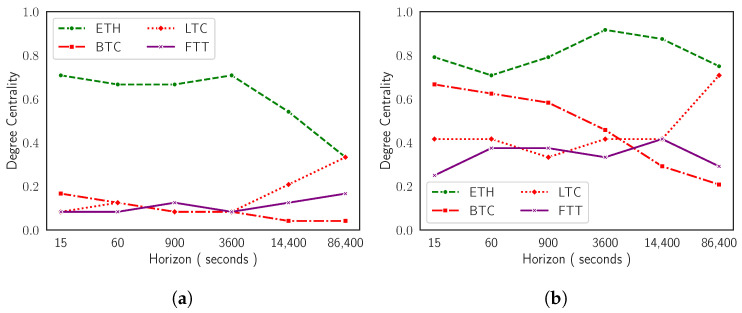
Degree centrality computed on MST (**a**) and on TMFG (**b**) as a function of time sampling resolution. Results on the TMFG highlight the switch in the reference roles of mainstream cryptocurrencies.

**Figure 6 entropy-24-01548-f006:**
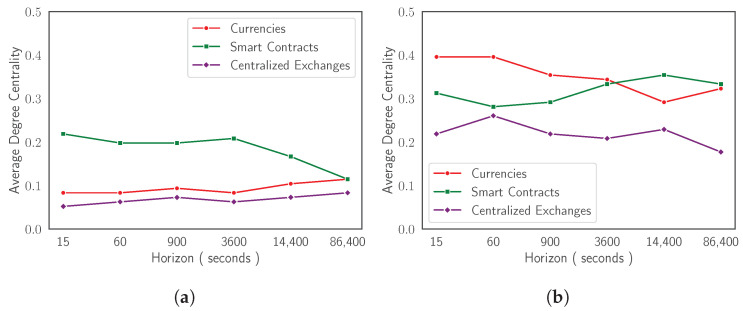
Group degree centrality computed on MST (**a**) and on TMFG (**b**) for Currencies sector, Smart Contracts sector, and Centralized Exchanges sector. Group degree centrality of a set of nodes is defined as the fraction of non-group members connected to group members. Sectors are defined following the taxonomy by [42].

**Table 1 entropy-24-01548-t001:** List of the 25 cryptocurrencies analysed in this paper. For each asset, the name, the symbol, the market capitalization at 29 March 2022 and the corresponding sector according to the taxonomy proposed by [42] is reported. There is no consensus on a unique mapping between cryptocurrencies and sectors. The chosen taxonomy is the one adopted by one of the main DCEs: Kraken [28]. Looking at the market capitalization column, it is worth noting that the least capitalized asset is Cream ($31.68M), while the most capitalized one is Bitcoin ($903B). Sectors’ grouping is balanced. Cryptocurrencies being the only representative of a specific sector are grouped together in analyses reported in Appendix A and Appendix B.

Cryptocurrency	Symbol	Capitalization	Sector
Aave	AAVE	$2.47B	Lending
Bitcoin Cash	BCH	$7.13B	Currencies
Binance Coin	BNB	$72.17B	Centralized Exchanges
Bitcoin	BTC	$903B	Currencies
Cream	CREAM	$31.68M	Lending
Ethereum	ETH	$412B	Smart Contract Platforms
FTX Token	FTT	$7.11B	Centralized Exchanges
Helium	HNT	$2.78B	IoT
Huobi Token	HT	$1.46B	Centralized Exchanges
Hxro	HXRO	$129M	Centralized Exchanges
Litecoin	LTC	$9.11B	Currencies
Polygon	MATIC	$13.21B	Scaling
Maker	MKR	$2.10B	Lending
OMG Network	OMG	$818M	Scaling
PAX Gold	PAXG	$609M	Stablecoins
THORChain	RUNE	$3.96B	Decentralized Exchanges
Solana	SOL	$36.09B	Smart Contract Platforms
Serum	SRM	$458M	Decentralized Exchanges
SushiSwap	SUSHI	$521M	Decentralized Exchanges
Swipe	SXP	$800M	Payment Platforms
TRON	TRX	$7.24B	Smart Contract Platforms
Tether	USDT	$81.37B	Currencies
Waves	WAVES	$5.77B	Smart Contract Platforms
XRP	XRP	$42.05B	Currencies
yearn.finance	YFI	$836M	Asset Management

**Table 2 entropy-24-01548-t002:** Average absolute correlation coefficient 〈|ρ|〉 and quantiles (25–75%) computed on the empirical correlation matrix **C**, on the links filtered by MST and on the ones filtered by TMFG at different time horizons. Statistical significance of the average correlation coefficient is represented though asterisks. *p*-values >0.05 are not marked. *p*-values ≤0.05 are marked as ${}^{*}$. *p*-values ≤0.01 are marked as ${}^{**}$. *p*-values ≤0.001 are marked as ${}^{***}$. The filtering power of the MST and TMFG is evident considering that the related mean correlation coefficients are always greater than the ones computed on the whole correlation coefficient matrix **C**. Results for both MST and TMFG are always robust across time horizons.

Δt	C			MST			TMFG		
	** 〈|ρ|〉 **	** 25% **	** 75% **	** 〈|ρ|〉 **	** 25% **	** 75% **	** 〈|ρ|〉 **	** 25% **	** 75% **
15	0.20	0.02	0.31	$0.35^{\;***}$	0.26	0.49	$0.31^{\;**}\;$	0.22	0.42
60	0.31	0.05	0.46	$0.47^{\;***}$	0.42	0.63	$0.44^{\;***}$	0.38	0.57
900	$0.44^{\;**}$	0.22	0.62	$0.60^{\;***}$	0.57	0.76	$0.57^{\;***}$	0.55	0.69
3600	$0.46^{\;**}$	0.28	0.63	$0.62^{\;***}$	0.59	0.76	$0.59^{\;***}$	0.56	0.70
14,400	$0.49^{\;*}\;$	0.38	0.65	$0.65^{\;***}$	0.64	0.77	$0.62^{\;***}$	0.58	0.72
86,400	0.48	0.38	0.62	$0.66^{\;**}\;$	0.64	0.77	$0.61^{\;**}\;$	0.57	0.72

**Table 3 entropy-24-01548-t003:** Percentage of links contained in empirical MST and TMFG at time horizon Δt with a bootstrap value larger than 95%. In the case of the MST, it is possible to notice how the robustness of the network structure decreases for coarser time sampling intervals. In the case of TMFG, on the contrary, the robustness is maintained across time horizons with low oscillations.

Δt	MST	TMFG
15	62.5%	28.9%
60	58.3%	37.7%
900	54.2%	36.2%
3600	58.3%	36.2%
14,400	41.6%	40.6%
86,400	25.0%	27.5%

## Data Availability

Data are accessible for free using the CCXT [41] Python Package.

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
