# Peer review of "Dependency Structures in Cryptocurrency Market from High to Low Frequency"

_entropy, 2022, doi:10.3390/e24111548_

Round 1
Reviewer 1 Report
The paper is very well written but, in my opinion, the authors should strengthen the motivation for their research from an economic and financial perspective, guiding the reader on the importance of analyzing the dependence structure across different timescales.
Other methodological choices, which are by nature arbitrary, also deserve some justification by the authors.
1. A major example is the choice of the power dissimilarity measure in eq. (4) could be critical to the results.
Have the authors performed the analysis with the standard Euclidean distance in eq. (3)? It would be interesting to check whether the results are robust to the choice of these correlation-based distances. The authors should address these points and give a motivation, which of course depend on the application they have in mind for the analysis, why a negative correlation should be considered equally important to a positive correlation of same size.
2. Minor examples are: the number of considered cryptocurrencies; the selected cryptocurrencies.
Specific and Minor comments:
- Since cryptocurrencies are traded 24h a day, the authors should specify want they mean with “closing price” (line 166)
- Figures 1 and 2 caption, ‘top k-central nodes are labeled’ but the value of k is not assigned explicitly
- Table 2 caption: the authors report (asterisks) the significance of the average correlation; which test is applied in order to obtain the correlation significance? The authors refer in the text to Appendix C which only contains pictures of the correlation distribution. I think that this point is not very clear and should be better explained.
Reviewer 2 Report
The paper “Dependency structures in cryptocurrency market from high to low frequency” explores and validates the Epps effect among cryptocurrencies for time scales spanning from 15 seconds to one day. Sparse representations of the correlation matrix are considered with the help of Minimum Spanning Trees and Triangulated Maximally Filtered Graphs, which confirm the Epps effect. The paper is well written, interesting and leads to intuitive conclusions. I only have minor remarks, which follow.
1- In equation 2, the correlation coefficient is introduced, with a T which is not defined. It is in fact not clear whether this correlation, for a given time scale of log-returns, is calculated for overlapping time increments or not. Said differently, the sum in equation 2 is not very clear as soon as Delta t does not appear explicitly. First, Delta t should appear in the log-return: so prefer x_c(t,\Delta t) instead of x_c(t). Second, is the sum in equation 2 something like \sum_{n}(x_i(n\Delta t,\Delta t)-\mu_i)(x_j(n\Delta t,\Delta t)-\mu_j) (non-overlapping increments) or like \sum_{n}(x_i(n \tau,\Delta t)-\mu_i)(x_j(n \tau,\Delta t)-\mu_j) (overlapping increments), where tau is the finest time step in the dataset (tau=15 sec. I guess)?
2- In Table 2, the average correlation coefficient is displayed but the average absolute correlation coefficient would be more relevant, as suggested by the choice made by the authors in equation 4. Having said that, negative correlations are very rare according to Figure 3, so that the suggested modification should not change a lot the conclusion. It is more about a concern for the consistency of the method.
